# Aorticoesophagal Fistula Combined with Upper Gastrointestinal Bleeding after Endovascular Dissection of Thoracic Aortic Aneurysm

**DOI:** 10.3390/diagnostics13010040

**Published:** 2022-12-23

**Authors:** Yongwei Xu, Ran Li, Kangwei Zhang, Shuchang Xu

**Affiliations:** 1Songjiang Hospital Affiliated to Shanghai Jiaotong University School of Medicine (Preparatory Stage), Shanghai 200065, China; 2Department of Gastroenterology, Tongji Hospital, Tongji University School of Medicine, Shanghai 200065, China; 3Department of Endocrinology, Tongji Hospital, Tongji University School of Medicine, Shanghai 200065, China; 4Department of Radiology, Tongji Hospital, School of Medicine, Tongji University, Shanghai 200065, China

**Keywords:** aortoesophageal fistula, thoracic aortic aneurysm, thoracic endovascular aortic repair, upper gastrointestinal bleeding

## Abstract

Aortoesophageal fistula (AEF) is a relatively rare and potentially fatal disease. Secondary AEF is rare but is associated with serious complications and high mortality. There are rare cases of esophageal mediastinal fistula after descending aortic aneurysm stent implantation. We report the case of a 76-year-old man who had upper abdominal distension, without obvious inducement, for 3 months and felt fullness after a meal, accompanied by anorexia. A chest computer tomography (CT) examination of the abdomen was performed with the outside hospital. Descending thoracic aortic aneurysm was discovered and was treated with stent implantation. The patient was transferred to our hospital to continue treatment, mainly because of an esophageal mediastinal fistula. Finally, the thoracic aortic aneurysm was diagnosed as AEF after stent implantation, combined with the diagnosis of upper gastrointestinal bleeding. We hope that, through this case, we can explain the possible causes of bronchial mediastinal fistula after stent implantation of descending aortic aneurysm and the mechanism of upper gastrointestinal bleeding.

Aortoesophageal fistula (AEF) is a rare disease with a very high fatality rate. AEF can be either primary or secondary to aortic homograft implantation. It can be caused by thoracic aortic aneurysm, foreign body ingestion, oesophagus malignancy, aortic injury, penetrating aortic ulcer rupture, thoracic aorta Aneurysm aortic stent placement (TEVAR) and other causes [1]. The incidence of AEF after aortic aneurysm stenting was 1.9% [2]. TEVAR has become a common treatment method for aortic aneurysms, of which endoleak is the main complication [3], and from which the biggest harm is the continuous enlargement or even rupture of the false lumen.

Surgical treatment is the main treatment for descending thoracic aortic aneurysms [4]. TEVAR is a minimally invasive and generally excellent modality of treating descending thoracic aortic aneurysms. However, paraplegia, stroke and occasionally AEF may be several complications of this [5]. AEF post-TEVAR is a rarely reported entity. Once it happens, it is a devastating and usually fatal condition. Treatment options are very restricted, as these patients are often not fit for complex surgery. Conservative management outcomes are almost always fatal due to recurrent haemorrhage or chronic infection and mediastinitis [6].

The most common and dangerous complication seen during treatment is massive haemorrhage [7]. Under the natural course of the disease, most patients will eventually die from massive haemorrhaging caused by an aneurysm rupture, and intraoperative bleeding is also a main reason for surgical death. An aortic oesophagus fistula is one of its rare complications.

This patient developed an esophageal mediastinal fistula after stent implantation for descending thoracic aortic aneurysm and was transferred to our department for treatment after a complicated infection. Finally, upper gastrointestinal bleeding occurred.

This case report aims to analyze the possible mechanisms leading to AEF. Furthermore, we hope that our report can supplement clinical data regarding AEF and provide a reference for clinicians to improve the diagnosis, treatment, and prognosis of this rare condition.

We reviewed the case of a 76-year-old male patient. He was diagnosed three months ago with upper abdominal distension without any predisposing factors, accompanied by anorexia, chest tightness, nausea, no vomiting, no abdominal pain and fever. Chest CT showed a descending aortic aneurysm with intramural hematoma. The small ulcer was treated with endovascular stent implantation for a thoracic aortic aneurysm, and the patient was discharged after improvement. After the patient was transferred to our department, the CT examination of the chest was perfected to indicate esophagus mediastinal fistula (Figure 1). After admission to the hospital, the patient’s body temperature slowly increased after admission (Figure 2). And the blood routine showed a decrease in HB. WBC, N, RBC, and PCT also change in volatility (Figure 3).

After the patient’s infection was controlled, and the endoscopy was perfected, we found that there was a fistula about 4 × 4 mm in the middle of the patient’s oesophagus (Figure 4A). According to the results of the gastroscopy, the esophageal mediastinal fistula was treated by clamping the esophageal mediastinal fistula with a titanium clip under the endoscope (Figure 4B). After the patient vomited blood, another enhanced CTA examination of the chest showed that there was an exudation of contrast agent around the stent. The possible cause of intermittent hematemesis was the formation of the patient’s oesophagus mediastinal fistula, and the mediastinum continued to be infected. Therefore, we considered that the patient to have invaded the descending thoracic aorta due to repeated infections of the mediastinum, causing continuous and slow bleeding (Figure 5). By carefully observing the patient’s chest CTA, it was found that there was blood oozing out of the descending thoracic aorta (Figure 6). However, according to the position of the calcification on the blood vessel wall, it was found that the blood vessel wall and the stent line formed a gap. Additionally, it was found that the oesophagus fistula was repaired. It can be seen after the treatment of the oesophagus fistula. Therefore, we also considered the possibility of a rupture of the peritoneal stent after treatment with the descending thoracic aorta peritoneal stent.

Among thoracic aortic aneurysms, the most common is descending thoracic aortic aneurysms. This is because atherosclerosis, trauma, bacterial infection and arterial necrosis may be the main causes of descending thoracic aortic aneurysms [8]. Endovascular stent graft implantation procedures are performed in patients with aneurysms of the aorta or other large vessels. The goal of the procedure is to preserve vessel patency and prevent the aneurysm from rupturing.

AEF is one of the main complications of TEVAR. Prosthesis infection, compression, ischemia, local inflammation and subsequent tissue necrosis are usually possible causes of AEF [5]. Grabenwöger M et al. studied relieved that secondary AEF can develop late after thoracic aortic surgery in up to 1.7% of patients [9].

Chiesa et al., believe that excessive stent size after TEVAR may be one of the important mechanisms of AEF formation. Most AEF patients have more than 20% or more proximal stents. These may lead to the deterioration of the arterial wall and migration of the stent graft, and could cause aortic aneurysm enlargement and bleeding [10]. Most AEFs are manifested by the Chiari triad. This sees chest pain first, followed by precursor arterial haemorrhage, before rapid and fatal haemorrhage occurs after some time, with time intervals ranging from several hours to several days [11].

According to the patient’s condition, a possible mechanism of the patient’s oesophagus mediastinal fistula is that the patient may have a stent implanted in the descending thoracic aortas. Then due to the tension of the stent, along with the pulsation of the blood vessel, the oesophagus constantly rubs against the oesophagus, resulting in a fistula. After the patient’s condition is stable, he actively undergoes endoscopic repair of the oesophagus mediastinal fistula.

Many treatments have been used for AEF after TEVAR, and these are mainly conservative and surgical treatments [12]. There are few kinds of literature about AEF after TEVAR. Akashi et al., reported that conservative treatment often has fatal consequences and is not recommended [13].

Uno et al., found that another TEVAR operation or combined oesophagus stent implantation can isolate the aortic blood flow and achieve the goal of controlling bleeding to treat AEF. However, after the implant is deployed, some patients are at risk of uncontrollable mediastinal infection and death [14]. Aortic replacement, combined with esophagectomy and debridement at the same time or in stages, is an effective treatment. However, complications such as haemorrhage and sepsis may occur [15,16]. Open surgery remains the treatment of choice in cases of preexisting vascular reconstruction or manifest infection. Endovascular techniques offer a viable solution in significantly comorbid patients or in patients presenting with acute, life-threatening bleeding [17]. The treatment of aortoenteric fistulas is associated with a high perioperative mortality rate [18].

In this case, AEF was diagnosed and accompanied by upper gastrointestinal bleeding. Sealing the endoleak and combining the oesophagus stent is recommended to prevent digestive juice from entering the mediastinum, control mediastinal infection, and provide opportunities for later treatment. The patient and his family refused the operation and asked to be discharged automatically. If this case is operated on in time, it may delay or prevent the occurrence of AEF, thereby increasing the long-term survival rate.

In conclusion, while TEVAR is a treatment option for thoracic aortic pathology, it does have limitations, such as anatomical problems. Enhancing overall understanding and individualized diagnosis and treatment measures are the keys to improving the survival rate of AEF after TEVAR. By reporting this case, we hope to raise awareness among physicians of the importance of pursuing an early diagnosis and identifying complications, factors which will improve clinicians’ understanding of AEF complications in TEVAR, enrich clinical knowledge, and provide suitable treatment options for future clinical reference.

## Figures and Tables

**Figure 1 diagnostics-13-00040-f001:**
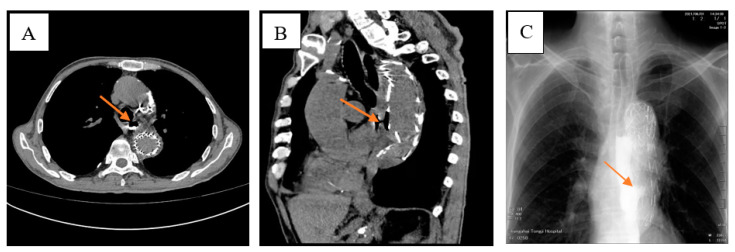
The chest CTA (computed tomography angiography) examination in the outside hospital revealed an esophageal mediastinal fistula and the patient was transferred to our hospital for treatment. Examination before admission of the patient revealed an oesophagus fistula. After the patient was transferred to our department, the CT examination of the chest was perfected to indicate an esophageal mediastinal fistula (**A**,**B**). An oesophagus lipiodol angiography was performed, and it was seen that the left edge of the middle oesophagus was approximately equal to the 8th thoracic saccular projection and diffused to the left posterior edge (**C**). It was considered that the oesophagus fistula may be large.

**Figure 2 diagnostics-13-00040-f002:**
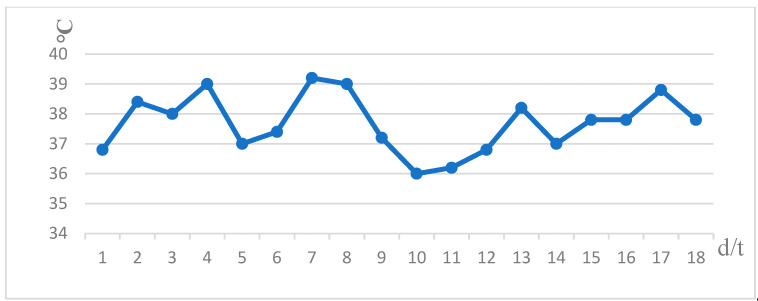
After admission to the hospital, the patient’s body temperature slowly increased.

**Figure 3 diagnostics-13-00040-f003:**
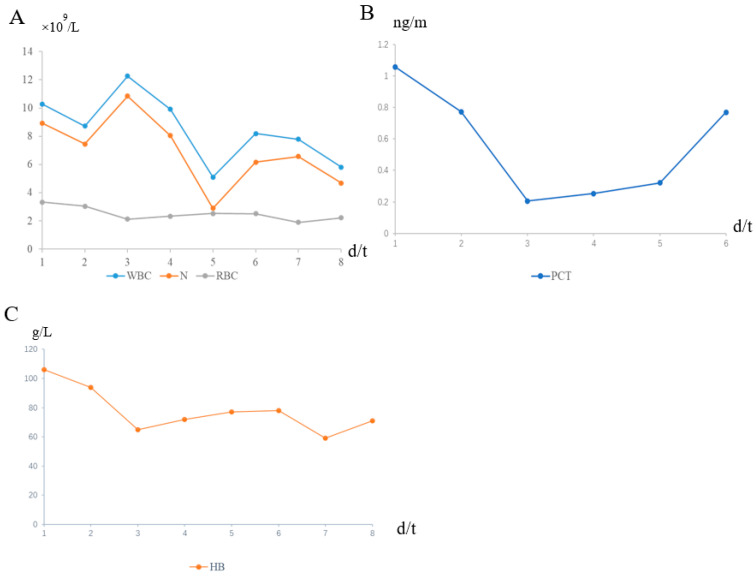
The patient developed intermittent hematemesis and the infection worsened. The patient’s blood routine examination showed that HB was lowered. After each blood transfusion treatment, the patient’s HB increased again (**C**). After the patient’s hematemesis was controlled, the re-examination under the endoscopy was performed again, and no bleeding points were found. Then, a series of inflammatory indicators were checked on the patient. The trend of changes in white blood cells (WBC) and neutrophils (N) after admission is basically consistent (**A**). After the patient was admitted to the hospital, the procalcitonin (PCT) increased significantly. After active treatment, the PCT improved significantly, but due to the continuous infection, the patient’s PCT testing increased again (**B**). The degree of changes in the red blood cell (RBC) to the haemoglobin (HB) in (**A**), after the patient is admitted, is mutually consistent (**C**). According to patients’ blood routine test results, acute infection was considered, and anti-infection was actively given for treatment. After anti-infective treatment, blood routine, PCT and other detection indexes were significantly decreased. During the subsequent course, the patient was infected repeatedly. WBC, reference range (3.5 − 9.5) × 10^9^/L; N, reference range (1.8 − 6.3) × 10^9^/L; RBC, reference range (4.3 − 5.8) × 10^12^/L; PCT: reference range <0.05 ng/mL; HB, reference range (130–175) g/L).

**Figure 4 diagnostics-13-00040-f004:**
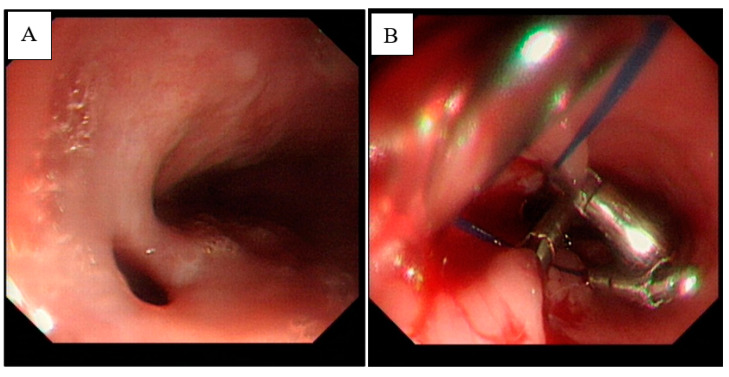
The chest CT examination of the patients before admission showed the formation of an esopha-geal mediastinal fistula. The inflammatory indicators increased, indicating the possibility of oe-sophagus mediastinal fistula infection. Next, by inducing fasting conditions, we reduced the risk of infection. After the patient’s infection was controlled, and the endoscopy was perfected, we found that there was a fistula about 4 × 4 mm in the middle of the patient’s oesophagus (**A**). According to the results of the gastroscopy, the esophageal mediastinal fistula was treated by clamping the esophageal mediastinal fistula with a titanium clip under the endoscope (**B**).

**Figure 5 diagnostics-13-00040-f005:**
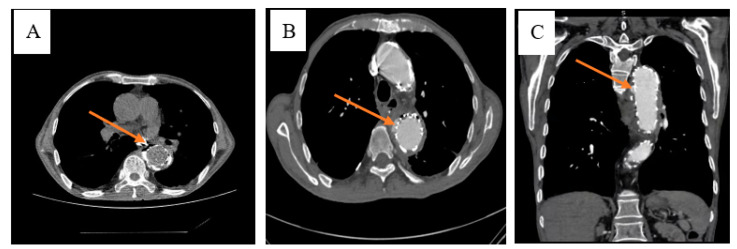
To clarify the cause of the patient’s gastrointestinal haemorrhage, the thoracic aortic CTA ex-amination revealed changes after the aortic arch-descending aortic stent implantation and a little contrast agent leaked from the outer edge of the descending aortic stent at the level of the flat chest 6–7 vertebral bodies. We saw the obvious changes after the titanium clip repair (**A**), and at the same time, there was blood leakage outside the descending thoracic aorta (**B**,**C**).

**Figure 6 diagnostics-13-00040-f006:**
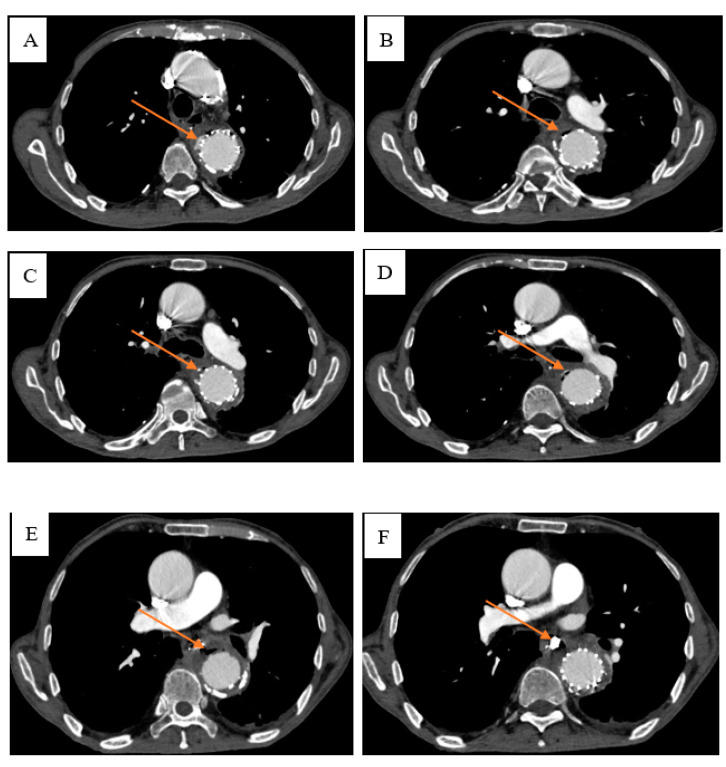
(**A**) Imaging findings of extravasation of blood outside the descending aorta. (**B**) Chest degradation of the aortic vascular wall. (**C**). Check the aortic blood vessel wall calcification stove. (**D**). The gap between the thoracic degradation of the aortic wall blood vessel wall and the membrane bracket. (**E**,**F**) Imaging findings after treatment of esophageal mediastinal fistula.

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
