# Peer review of "Aorticoesophagal Fistula Combined with Upper Gastrointestinal Bleeding after Endovascular Dissection of Thoracic Aortic Aneurysm"

_diagnostics, 2022, doi:10.3390/diagnostics13010040_

Round 1
Reviewer 1 Report
This case report of Aortoenteric Fistula (AEF), which is one of these rare complications, will contribute to the scientific literature. both diagnostic approach and treatment planning are very interesting.there may be some word mistakes in the article : ''oesophagal''
Round 1
Response to Reviewer 1 Report
This case report of Aortoenteric Fistula (AEF), which is one of these rare complications, will contribute to the scientific literature. both diagnostic approach and treatment planning are very interesting. There may be some word mistakes in the article: ''oesophagal''
Author Response
Thank you for your comments!
We are apologizing for this mistake, thank you. The wrong spelling has been revised in the article.

Reviewer 2 Report
- very interesting topic for all readers. pls pay attention to the following publications
Sachsamanis G et al:Midterm Results after Open Surgical and Endovascular Management of Arterioureteral Fistula.Ann Vasc Surg. 2021 May;73:280-289 and Treatment of Secondary Aortoenteric Fistulas Following AORTIC Aneurysm Repair in a Tertiary Reference Center Oikonomou K et al J Clin Med. 2022 Jul 29;11(15):4427.
Round 1
Response to Reviewer 2 Report
very interesting topic for all readers. pls pay attention to the following publications
Sachsamanis G et al:Midterm Results after Open Surgical and Endovascular Management of Arterioureteral Fistula.Ann Vasc Surg. 2021 May;73:280-289 and Treatment of Secondary Aortoenteric Fistulas Following AORTIC Aneurysm Repair in a Tertiary Reference Center Oikonomou K et al J Clin Med. 2022 Jul 29;11(15):4427.
Author Response
Thank you for your comments!
After reading these two articles, I found out that open surgery remains the treatment of choice in cases of preexisting vascular reconstruction or manifest infection. Endovascular techniques offer a viable solution in significantly comorbid patients or in patients presenting with acute, life-threatening bleeding. But, treatment of aortoenteric fistulas is associated with a high perioperative mortality rate.
Surgical surgery is a method for treating esophageal longitudinal fistulas. These two articles mention the cause of the death of patients after surgery, which has a lot to do with the stability of the patient's blood flow dynamics. Therefore, I quoted the article and hoped to provide a certain reference to surgery.
The articles were quoted [18], [19].
[18] Sachsamanis G, Pfister K, Kasprzak PM, et al. Midterm Results after Open Surgical and Endovascular Management of Arterioureteral Fistula. Ann Vasc Surg. 2021, 5, 280-289.
[19] Oikonomou K, Pfister K, Kasprzak PM, et al. Treatment of Secondary Aortoenteric Fistulas Following AORTIC Aneurysm Repair in a Tertiary Reference Center. J Clin Med. 2022, 15, 4427.
